# Distributed Detection of Malicious Android Apps While Preserving Privacy Using Federated Learning

**DOI:** 10.3390/s23042198

**Published:** 2023-02-15

**Authors:** Suchul Lee

**Affiliations:** Department of Data Science, Korea National University of Transportation, 157, Cheoldobangmulgwan-ro, Uiwang-si 16106, Gyeonggi-do, Republic of Korea; sclee@ut.ac.kr; Tel.: +82-31-460-8860

**Keywords:** federated learning, deep learning, privacy, malicious android app detection, FedAvg algorithm, stream order imaging, independent and identically distribution (IID)

## Abstract

Recently, deep learning has been widely used to solve existing computing problems through large-scale data mining. Conventional training of the deep learning model is performed on a central (cloud) server that is equipped with high computing power, by integrating data via high computational intensity. However, integrating raw data from multiple clients raises privacy concerns that are increasingly being focused on. In federated learning (FL), clients train deep learning models in a distributed fashion using their local data; instead of sending raw data to a central server, they send parameter values of the trained local model to a central server for integration. Because FL does not transmit raw data to the outside, it is free from privacy issues. In this paper, we perform an experimental study that explores the dynamics of the FL-based Android malicious app detection method under three data distributions across clients, i.e., (i) independent and identically distributed (IID), (ii) non-IID, (iii) non-IID and unbalanced. Our experiments demonstrate that the application of FL is feasible and efficient in detecting malicious Android apps in a distributed manner on cellular networks.

## 1. Introduction

Currently, smartphones have become some of the most common and popular electronic devices due to the rapid development of cellular wireless communication technology and mobile operating systems. According to Gartner’s 2022 report, the market share of Android OS, which is adopted by most smartphones, except for Apple’s iPhone (iOS), has steadily increased and has recently reached nearly 90% [1]. Android’s rapid growth is driven by openness; Android apps can be developed and distributed by anyone as long as they are adopted by the market.

Unfortunately, Android’s openness does not provide pure functionality. The nature of open platforms involves accelerating the emergence of new malicious Android applications. According to Kaspersky Lab’s Q2 2022 IT Threat Report [2], 5.5M mobile malware, adware, and riskware attacks were blocked, of which, 405,000 were installed on clients. It contains approximately 55,000 mobile banking Trojans and 4K mobile ransomware.

Conventional malicious application detection is based on unique signature matching based on static/dynamic analysis of malicious code [3]. However, this signature matching-based detection technique can be easily defeated by techniques such as application code obfuscation, manifest cheats, and dynamic code loading. To make matters worse, the rapid proliferation of malicious applications makes it nearly impossible to quickly collect unique signatures for new malicious applications [4].

Recently, in various computer science fields, such as communication and information security, machine learning (ML) technology has been widely used as a tool to solve existing computing problems through large-scale data learning. Efficient and accurate training of ML models depend almost entirely on the quality of the training data and the computing power of the server performing the training. Therefore, in most cases, it is implemented to maximize efficiency in terms of power/time by collecting data from various clients in a high-performance centralized data server and training the model through the server’s powerful computing resources [5].

However, centralizing all raw data to train and test ML models inevitably faces challenges, such as (i) resource constraints, (ii) security and privacy concerns, and (iii) securing communication bandwidth to transmit raw data [6]. In particular, in the case of a cellular network, transmitting a smartphone user’s application installation and usage status, website access record, contact information, multimedia, etc., to a central server not only consumes bandwidth but also causes serious personal information leakage. In this case, the user may not cooperate with the process of collecting raw data from the central server to protect personal information. Even worse, malicious users could also exploit this raw data collection process to compromise data integrity, intentionally preventing effective ML model training [7,8].

In a context where privacy and data integrity are important, federated learning (FL) has been highlighted as a distributed collaborative ML paradigm [9]. In FL, instead of sending raw data to a central data server to train global ML models, clients use local data to train each local ML model in a distributed fashion. Trained local ML model parameters are sent to a central server for integration into the global ML model. A unified global model is deployed to each client and is used to augment the local model in the next local training process. By repeating this process, each client participates in a decentralized global model training process. In FL, data privacy is guaranteed by design because the client’s raw data are not shared with the outside, and the client only needs to transmit fixed ML model parameters regardless of the size of the raw data, so it is free from the problem of securing communication bandwidth.

Despite the various strengths of FL, there are several issues to consider when applying FL to real-world environments. The exploration of how to train an accurate global ML model under a statistically unbalanced distribution of learning data across clients is the most universal but important research topic [10,11,12,13]. Even without direct transmission of raw data, the communication cost for model updates is high because many clients are involved and each client sends a massive number of parameters of deep neural networks. In order to reduce the communication cost, a method of transmitting parameters in a compressed form [14] or minimizing the number of model updates [15] has been proposed. Considering that deep neural networks are vulnerable to adversarial attacks in the domain of image classification, the problem is exacerbated in FL, where the attack target increases significantly. Accordingly, in the information security field, discussions are actively underway to secure resilience against such adversarial attacks [8,16,17].

In this paper, we conducted an experimental study to investigate the feasibility and efficiency of a malicious Android app detection method through FL under the practical challenges raised above. To this end, we first propose an FL model that simultaneously considers computing/communication resource limitations and privacy issues in a cellular network environment. In order to extract features required for deep learning, we present an imaging technique for malicious Android app samples in the form of APKs. Finally, we perform extensive FL learning experiments using real smartphone data under three data distributions; FL over independent and identically uniform distributed (IID) local data, FL with non-IID local data, and FL with non-IID and imbalanced local data. The experiments include explorations of the impacts on client availability, local computing costs, and communication overhead (between the center and local). The main findings obtained through the experiments are summarized as follows.

Application of privacy-preserving FL is feasible in detecting malignant Android apps in a distributed fashion in actual cellular networks.Fewer global training rounds for the FedAvg algorithm [15] typically translates into increased efficiency.However, we could not observe any clear correlation in training efficiency with changes in client availability and/or local training intensity. Rather, the biggest factor affecting efficiency is the communication overhead of updating model parameters.

The remainder of this paper is organized as follows. Section 2 reviews related work. Section 3 presents the system model and briefly describes the parameter update process and FedAvg. Section 4 describes the experimental setup in detail, and in Section 5, the experimental results and corresponding analysis are presented. Section 6 discusses various practical issues. Finally, Section 7 concludes this paper.

## 2. Related Work

For the past several years, ML technology research has been conducted, such as deep learning used to detect malicious codes. In general, ML technology must go through a process of learning an ML model that reflects the statistical characteristics of the target through a large amount of training data. The detection performance of a trained model depends on the process of identifying and extracting features, in which case domain knowledge of the data is important. This process, called feature engineering, is a fundamental part of ML applications and is an important factor affecting the performance of ML algorithms. Depending on the kind of features extracted, related studies fall into two categories: signature- or behavior-based techniques.

### 2.1. Signature-Based Detection

Signature-based detection is frequently used in the traditional information security field, which emphasizes a clear correlation between malicious code and signatures. It detects malware using known malicious code signatures. For this, it is necessary to search for the signature unique to the malicious code through static analysis. Therefore, signature-based methods can detect known malicious apps with high accuracy. However, detection of zero-day vulnerabilities is fundamentally impossible because detection is possible only by knowing the unique signature of the malicious code. In addition, attackers can easily bypass detection by obstructing the discovery of unique signatures with techniques, such as code obfuscation. Reference [18] introduced the concept of data mining for malware detection and proposed a detection technique that applied the Ripper algorithm to three static features of the PE (portable executable), string, and byte sequence. In the study by Kong et al. [19], the function call relationship of malware was expressed as a graph, and a graph-based malware detection technique was proposed. In this technique, the distance between malware is quantified through the ensemble value of the model, and malware is effectively classified based on this. Li’s study [20] was conducted on a mobile environment, and based on the statistical similarity of API calls and code structures in Android APKs, malware operating in the mobile environment is classified.

The ultimate way to find unique signatures of malicious code using static analysis is to utilize a technique called reverse engineering. In general, since it is very difficult to obtain the source code of the malicious code that exists in the form of a PE file, the malware analyst disassembles the malicious code binary and applies feature engineering at the assembly level. Reference [21] proposed a method utilizing opcode sequences to construct vector representations of executable files. Many studies have been conducted to classify malware by converting the malware binary into an image [8,22,23,24]. Specifically, Reference [22] proposed a method of classifying malware by applying simple hashing to the opcode sequence. Moreover, in [23], the malware was classified by visualizing the malware binary as a grayscale image and applying the K-nearest neighbor algorithm. Reference [24] proposed a method for detecting malware through image entropy quantification. Reference [8] showed that the Convolutional Neural Network (CNN)-based malware detection algorithm can be neutralized by the adversarial examples (AEs) generated using Generative Adversarial Networks (GAN), and suggested a robust training method for a deep learning model resilient to these AEs. Reference [8] is complementary to this paper in that it provides a countermeasure against adversarial attacks to the proposed method, which is discussed artificially in Section 6.

### 2.2. Behavior-Based Detection

Signature-based techniques are usually combined with static analysis to find malware-specific signatures. However, in this way, it is impossible to capture unique signatures generated only during the execution of malware, such as network packets, API calls, opcode sequences, etc. Even static analysis of obfuscated or encrypted malware is fundamentally impracticable. The collection of these features must be accompanied by the execution of malware, which is performed through artificial and/or bystander execution of malicious codes in a virtual machine environment thoroughly isolated from the external network, e.g., the internet. That is why these kinds of features are called behavioral features, so detection techniques based on these features are called behavior-based detection.

Bayer et al. [25] proposed a method of clustering PE files based on behavior and detecting/classifying malware. Reference [26] created a graph based on the opcode trace and presented a malware detection technique through graph analysis. Fujino et al. [27] presented a measure to quantify the similarity of malware based on the API function calls, and a detection method applying the non-negative matrix factorization technique.

Various practical applications using dynamic analyses have been proposed due to their advantages of finding malware features that are inherently impossibly analyzed with static analyses. However, clever attackers have evolved to circumvent malware detection techniques built upon the dynamic analysis. For example, if malware is designed to be latent for a certain period of time before it actually performs malicious behavior, sandboxes can never detect malware before it becomes active. As another example, some malware can only be activated through a network connection and, therefore, in a sandbox operating in isolation from the internet, malware of this kind is not activated forever. As such, the dynamic analysis does not always guarantee the extraction of malware features.

In the field of federation optimization, several studies have been conducted on the convergence and efficiency of FL under statistically non-IID and imbalanced training data [10,11,12,13]. Reference [12] presented a mathematical definition of the heterogeneity of data. In [13], the conditions for ensuring convergence and the asymptotic bound required to reach the optimum were derived through mathematical analysis. Reference [15] experimentally showed the dominance of the communication costs for model updates and proposed the FedAvg algorithm, which opened up the door to one of the federated optimization studies. In [13], it was mathematically shown that the FedAvg algorithm can ensure convergence by choosing an appropriate sampling method. FedPer [28] uses the concept of multi-task learning to deal with statistical heterogeneity. The personalization layer is trained by mutually sharing the trained base layer using the FedAvg algorithm. FedProx [29] adds a proximal term to the local loss function to prevent the local and global models from being too different in heterogeneous network environments. However, this study is differentiated in that it experimentally analyzed the feasibility and efficiency of FL application in the actual cellular environment from various angles.

## 3. A Privacy-Preserving Cross-Silo Federated Learning Framework

### 3.1. System Model

In this paper, as shown in Figure 1, we assume a cross-silo FL model [30] for the purpose of detecting malicious Android apps in cellular networks. In the proposed FL model, the (cloud) data server acts as a central entity. The central entity cooperates with edge entities to train the global ML model and update the local ML model parameter values. In our cross-silo FL model, an edge entity means some representative clients or a base station (BS). In cellular networks, most terminals are smartphones, and since the number of users is very large, it may seem reasonable to apply the cross-device FL model. However, from a global training perspective in FL, users’ personal data are often not enough to train local ML models, and more importantly, model updates require excessive communication bandwidth between the central server and clients. It is unnecessary and unwelcome for individual smartphone users. Therefore, in our model, the BS with relatively better computing and communication performance compared to the client becomes the edge entity (silo 2 in Figure 1). Or, it is assumed that clients form a coalition and some of them act as the edge entity on behalf of the coalition (silo 1 or 3 in Figure 1).

**Preserving privacy within the coalition**: By forming a coalition between the edge entity and the client in the proposed model, the personal information leakage problem seems to become an issue again. In the proposed FL model, it is assumed that the local differential privacy (DP) mechanism [31,32] is applied in the process of collecting raw data within the coalition. The DP algorithm adds perturbation to raw data while maintaining statistical characteristics, making it completely different from the original raw data, thus preserving privacy. However, since deep learning operates based on the statistical characteristics of data, it is possible to learn models without performance degradation by properly designed perturbation.

### 3.2. Parameter Update

In this paper, using a controlled environment suitable for experiments, we examine the impact of client availability, local computing intensity, and data distribution across edge entities in depth. In our FL model, synchronous parameter update is performed every epoch time. We assume there are *K* edge entities over which the data are partitioned, with $D_{i}$ the set of indexes of data points on edge entity *i*, with $n_{i}=\left|D_{i}\right|$. In every update round, K×C edge entities are randomly selected out of *K* edge entities, and the central entity transmits the current global model state (i.e., current global model parameters) to the selected edge entities. Here, *C* can be viewed as an experimental parameter implying the availability of edge entities. Each edge entity trains a local ML model based on global model parameters (distributed from the central entity) and local data. It then sends parameter updates to the central entity. The central entity then aggregates the received updates and integrates them into the global ML model to complete synchronous parameter updates during one epoch.

### 3.3. Federation Algorithm

State-of-the-art methods for neural network (NN)-based ML algorithms typically use stochastic gradient descent (SGD) to train model parameters. Over the past years, several studies have been conducted to develop effective algorithms for applying SGD to FL [15,33]. In this paper, we apply the state-of-the-art FL algorithm, *federated averaging*(FedAvg) introduced in [15], and conduct an extensive experimental study to verify the effectiveness of Android malicious app detection in FL environments. Note that the main contribution of this paper is not to propose a new FL algorithm, but to experimentally examine the feasibility and efficiency of the FL technique in a real cellular environment. FL is a method of physically distributing the training process performed on a central data server, so the maximum performance that can be achieved is bounded by the performance of the centrally trained ML model.

**Baseline**: Chen et al. [34] presented a decent approach to a centralized setup, namely FedSGD, which outperforms previous asynchronous approaches. An algorithm that is mathematically equivalent to this method can be performed in a federated setting. In each round, we select K×C edge entities and compute the gradient of the loss for the local data. Therefore, *C* can be seen as controlling the global batch size, and when C=1, it is non-stochastic full-batch gradient decent, mathematically equivalent to FedSGD. Therefore, if the communication cost is not taken into account, the performance target of a model trained with FL is the same as that of FedSGD. The main difference between FedSGD and FedAvg is that FedSGD allows the client to train only one step before sending the updated weights to the server, so FedSGD is called ‘*baseline*’.

## 4. Experimental Setup

### 4.1. Experiment Environment and Dataset

The experimental dataset used in this paper is CICMalDroid 2020 [35]. The Maldroid dataset was collected from third-party sources, such as VirusTotal and Contagio blogs in the form of an APK, which is an Android app installation file from December 2017 to December 2018. It consists of a total of 17,341 samples, including both malignant and benign Android app samples. In Apple’s iOS, it is impossible in principle to install apps other than apps certified in the App Store without rooting (jailbreak). On the other hand, in the Android environment, security threats are inherent because users can freely install apps as they wish.

Our experiments were implemented utilizing the PyTorch framework and were performed on a MAC laptop with an Apple M1 Pro CPU and 32B RAM. The M1 laptop does not have a separate GPU installed other than the CPU’s built-in GPU. Therefore, the GPU acceleration using the built-in GPU is called MPS (metal performance shader) and has quite usable performance.

#### 4.1.1. Android App Imaging

**APK File Pre-processing**: Since the APK file is in the form of a ZIP compressed file, if it is converted to an image without pre-processing, there is a possibility that important features for ML model learning will be compressed and lost. In this paper, the following three files are extracted from the APK file and used.

AndroidManifest.xml: This file is the first file read when running the application. It stores application-essential information, such as components, hardware capabilities, and user rights.classes.dex: Dalvik opcodes compiled to be executable on the Dalvik virtual machine.resources.arsc: These are xml files compiled into binaries that are necessary for APK execution.

**Stream Order**: A CNN or MLP (multi-layer perception) network architecture is commonly used to apply deep learning to non-sequential data. Since this architecture is usually applied to training data in the form of images, we need to convert the Android app sample in the form of an APK to an image. We convert the APK file into an image file using the stream order (SO) method as follows. We concatenate the three files extracted in the APK file pre-processing process. Let *S* be the byte size of the merged file; *S* determines the size of the generated image in the form of a square. Moreover, 2 bytes (16 bits) are sequentially read from the merged file, and every 2 bytes read is converted into a number from 0 to 255, which is mapped to a grayscale image color. A total of S/2 readings are performed. Specifically, the size of an image is given as $\left\lfloor \sqrt{S/2}\right\rfloor \times \left\lfloor \sqrt{S/2}\right\rfloor$. So, the coordinates of the vertices of the square-shaped Android app image are (0,0), $\left(0,\left\lfloor \sqrt{S/2}\right\rfloor -1\right)$, $\left(\left\lfloor \sqrt{S/2}\right\rfloor -1,0\right)$, $\left(\left\lfloor \sqrt{S/2}\right\rfloor -1,\left\lfloor \sqrt{S/2}\right\rfloor -1\right)$. For example, if the read value is 0x00, 0xFF, 0x00, …the image coordinates (0, 0), (0, 1), (0, 2), …must be white (0x00), black (0xFF), white (0x00), …, respectively. Since the size of *S* is different for each Android app, the Android app image size is also different. Therefore, the final image is re-sized to 256 × 256. Figure 2 is a sample Android app image created using an APK file.

#### 4.1.2. Separation of Training/Testing Data

For fair learning and evaluation of the global FL model, 14,000 samples are randomly selected out of a total of 17,341 Android app samples and used as training data, and the remaining 3341 samples are used as evaluation data. Here, random selection is performed for each experiment (FL model learning instance), so the four types of malicious and non-malicious Android app samples vary in each experiment.

#### 4.1.3. Data Distribution over Edge Entities

The three data distributions (IID, non-IID, and non-IID—imbalanced across edge entities) in which the FL algorithm operates are mathematically described in Section 3.2. We implement actual data distribution by allocating Android app images to edge entities through the following sampling process. Note that we now have a total of 14,000 training samples.

**IID**: The training samples are shuffled, and then 14,000/K samples are allocated to each entity.**Non-IID**: First, we sort the data by label (android app type), divide them into 200 shards at a size of 70, and allocate 200/K shards to each *K* edge entity. As a result, most edge entities only have training samples for two classes of applications, this is a so-called pathological non-IID partition of the data. Note that IID and non-IID partitions are balanced.**Non-IID and imbalanced**: It is similar to non-IID. First, we sort the data by label and divide them, e.g., into 1400 shards at a size of 10. We allocate at least one shard and a maximum of $Shard_{max}$ shards to each *K* edge entity. Similar to non-IID, it constitutes a pathological non-IID partition of data, and there are many allocation methods that lead to sufficient imbalance.

### 4.2. FL Training Setup

#### 4.2.1. Deep Learning Networks

Our experiments are conducted using convolutional neural network (CNN) architecture, which is simple and widely used.

Convolution L1: 3 × 2 convolution with a 5 × 5 kernel, a stride of 1, ReLU activation.Max pooling L1: followed by convolution layer 1, with a 2 × 2 kernel and a stride of 2.Convolution L2: 6 × 16 convolution with a 5 × 5 kernel, a stride of 1, ReLU activation.Max pooling L2: followed by convolution layer 2, the same with max pooling layer 1.Fully connected L1: 59,536 input features are connected fully to 120 out features.Fully connected L2: 120 input features are connected fully to 84 out features.Fully connected L3: 120 input features are connected fully to 5 out features.Softmax: performing classification.

The CNN is a structure with two convolution layers connected to a 2 × 2 max pooling layer and three fully connected layers connected after the second max pooling layer. The last fully connected layer finally classifies images into five classes of Android apps using Softmax.

#### 4.2.2. FL Hyperparameters

Parameters were fine-tuned to provide optimal performance for the FL model on the CICMaldroid dataset and deep neural networks used in all experiments. Table 1 shows the federation hyperparameter values finally used in our experiments. We only control *C* and *E*. Here, the local batch size, *B*, is also a hyperparameter related to the training amount of edge entities (it increases when *B* decreases or *E* increases). In our experiments, the dataset is distributed among 100 edge entities; thus, the data that each edge entity can hold are restrictive. Experiments with *B* as a controlling variable are also restrictive. Therefore, we do not control *B* in our experiment, and are directly related to the federated optimization to explore their impacts on the performance of the trained FL model. Here, *C* means the ratio of edge entities participating in global training, i.e., the availability of edge entities, and *E* means the number of local training rounds, i.e., the degree of local training at edge entities. Moreover, we validated learning rate values over a sufficiently wide range and at a fine granular level. It is impossible to prove that a learning rate value of 0.01 is mathematically optimal, but experiments have shown that it converges quickly to the optimal point in most cases. Therefore, we use this value regardless of changes in *C* and *E* throughout the remainder of our experiments.

## 5. Experimental Results

### 5.1. Feasibility of Applying FL

Our initial experiments aim to show the *feasibility* of our FL-based distributed malicious Android app detection method. Figure 3 shows the performance of deep learning models trained using CNN. In this experiment, the FL hyperparameter values were set to C=0.1, B=10, and E=10. We observed that the three schemes converged almost the maximum performance after a relatively small number of training rounds, except for the most challenging training, with non-IID data distribution. This means that with FL, one can obtain a distributed trained model with comparable performance to a model trained on a central data server.

With pathologically partitioned non-IID data, the FL training process did not converge to target accuracy within one thousand rounds of training. Applying the sampling process described in Section 4.1.3 to form non-IID, assigning training data to edge entities, each edge entity is usually assigned two classes of Android app data. This is an attempt to train an entire deep neural network with the local training of edge entities that even observe a (biased) part of the overall data. It never converges to the global optima. We conjecture that, in the modeling process, the gradient descent cannot easily deviate from the local optima.

This claim is also supported by Figure 4, a graph of the change in training loss according to training rounds. The training losses for the non-IID data are unexpectedly close to the baseline, meaning that the FL trainer is already performing the best. Since the baseline scheme uses the entire data to train the ML model, it is surprising that the training loss graph comes closest to training with Non-IID using partial and biased data. Considering the characteristics of the SGD optimizer, we conjecture that the optimal point (although this may not be global optimal) search by gradient descent is effectively performed in a given convex data plane.

Interestingly, when the data are non-IID and disproportionately distributed among the edge entities, the test accuracy curve of the FL model becomes significantly close to that of FL under IID data distribution. This can be explained as follows; as described in the Section 4.1.3, if different amounts of non-IID data are allocated to each edge entity (of course, the edge entity to which each data shard is allocated is selected uniformly at random), some edge entities are allocated shards to contain all Android app class labels because the number of shards far exceeds the number of edge entities. In this case, it can be understood that the degree of non-IID [13] decreased while grouping non-IID data chunks, and this phenomenon was also reported in [12]. In federated optimization, the global loss is given as a weighted sum of local losses. During sampling, some edge entities are allocated more data shards, resulting in a lower degree of non-IID. These edge entities are likely to dominate other edge entities due to their relatively higher weight during the aggregation process.

The encouraging fact here is that the quantitatively imbalanced non-IID data distribution is most mimic to the real mobile environment. In cellular networks, the distribution of local data of a specific user is different from that of global data because it is based on the user’s use of a mobile device. In addition, some heavier users use certain services or apps far more than others depending on activity categories such as user interests or occupations, leading to non-IID and an imbalance of local training data between edge entities. Our experiments show that FL using FedAvg works fairly well for most realistic data distributions. *In conclusion, we show that FL is a very feasible solution for detecting malignant Android apps in a distributed fashion.*

### 5.2. Efficiency Gains through Distributed Computing

A second discussion of the experimental results involves the efficiency gains that can be achieved by distributing the computational load from the central entity to the edge entity. We investigated the number of training epochs required to achieve the target test accuracy, i.e., 0.91 for CNN. Moreover, we compute the wall-clock time required to achieve the above accuracy goal by considering the parallelism of local training proportional to the number of edge entities participating in global training and the communication time required for parameter updates. We assume a communication bandwidth of 54 Mbps, which is a fairly conservative value considering that the current cellular network has gone beyond the popularization of 5G and is approaching 6G.

Table 2 shows the results. Overall, training deep learning models in parallel increases efficiency; efficiency gains range from 1.5× to 4.27×. In FL, even if more edge entities participate in global training, it is predicted that the wall-clock time required for local training will not be significantly affected due to the parallelism of edge entities. Our experimental results, however, show that this is not always the case.

The number of parameters of the simple CNN in the experiment is 7.1M (See Section 4.2.1), and under the above-mentioned assumptions, updating the parameters of one edge entity takes approximately 8.09 s. Moreover, the central entity must receive updates from all participating edge entities in every round of aggregation. In fact, in experiments with C = 0.1 and E = 10, the local training time and parameter update of one edge entity take 10.59 s and 8.09 s, respectively. Here, since parameter update is not subject to parallel processing, communication overhead increases as more edge entities participate or central/edge entities have better computing power. In the same vein, the reduction in the number of training rounds to reach the target test accuracy is not significantly affected by an increase in *C*, thus reducing the efficiency gain (see Table 2, when C=0.3 speed-ups is less than 1).

*In summary, a reduction in training rounds in the FL setting usually leads to an increase in efficiency.* Given FL’s strengths in terms of privacy, it is quite attractive for smartphone users to unite (just install the app) to protect their privacy while detecting rogue Android apps installed on their phones. Still, the communication overhead due to global parameter updates has a significant impact on efficiency, which will be explored in more detail in the next subsection.

### 5.3. Impact of Hyperparameters

**The availability of edge entities**: In FL, as the availability of edge entities increases, the number of edge entities selected for the local training process increases. Physically, more smartphone users have increased their voluntary participation in the collaborative detection of malicious Android apps. Figure 5a,b show the results. As also observed in Section 5.2, an increase in *C* did not have a clear effect on the number of training rounds to reach the target test accuracy for all three data distributions used in the experiment. In addition, the (wall clock) time required for local training is independent of the number of edge entities participating in training, but the parameter update time is at least proportional to the number of entities.

**Increasing local training**: In FL, we can expect better fitness of the local model to the local data by increasing the number of training rounds on edge entities. If training is excessive compared to the data and model structure, you may run into the local overfitting problem. In our experiment, *C* was fixed at 0.1, and based on the fact that the network model converged to the target before and after about 200 training rounds, the experiment was conducted while changing the *E* value to 10, 20, and 30.

Figure 6a,b show the results. As in the experiment on the change in *C*, it was difficult to find out the specific effect of the increase in *E* on the number of training rounds. Since many edge entities are involved in global training, we conjecture that the fitness of each model to the local data is neutralized during aggregation. This claim is also supported by Figure 5b and Figure 6b, which will be discussed in detail in the following two paragraphs.

**The mitigation of fluctuation**: We observed significant fluctuations in both the test accuracy (Figure 3) and training loss (Figure 4) graphs. This oscillation is more evident in the loss graph; See Figure 5b and Figure 6b. In general, there are two reasons for severe oscillation in the test accuracy and training loss curves during centralized training. (i) If the local batch size (*B*) is too small, (ii) and if the sizes of the dataset and the training network are too different.

In FL, in addition to the factors mentioned above, the difference in fitness to the local model is a major factor in oscillation in the aggregation process. As the local models trained under non-IID data are predicted to be the most dissimilar, and the corresponding evidence is easy to be observed in Figure 5b and Figure 6b (schemes under non-IID data are plotted as blue lines.). Encouragingly, increasing local training (whether through increasing *C* or *E*) reduces the oscillation. Even under non-IID data distribution, the data are not completely different between the edge entities, which means that the convergence performance degradation due to non-IID data can be mitigated by increasing the local training intensity.

*We could not observe a clear correlation between changes in FL hyperparameters and the number of update rounds in the test accuracy and training loss curves. Training more locally, such as increased availability of edge entities or more computations on edge entities, does not directly affect efficiency gains, but the overhead for communication of update parameters does have a relatively large effect.* On the other hand, in Section 5.2, we observed speed-ups from a minimum of 1.5× to a maximum of 4.27× for most FL settings. However, on the graph shown in Figure 5a and Figure 6a, this tendency towards speedup cannot be asserted. We conjecture that the oscillation observed in FL’s global training process, i.e., parameter aggregation, is the basis for this phenomenon. When the deep learning model is trained to a certain extent, the training can be ended probabilistically by oscillation. However, this is not a completely random point in time, it is a point that is close to the target accuracy. This is why there are some state-of-the-art studies on reducing the communication overhead for parameter updates [14,15].

## 6. Discussion

### 6.1. Is 91% of Test Accuracy Sufficient?

Recently, AI-based technologies, such as deep learning, are actively underway in the information security field. Still, commercial vendors employ a rule-based detection method for their information security products, such as intrusion prevention systems (IPSes), web application firewalls (WAFs), and virus scanners. This is because the detection of malicious codes usually requires a conservative approach. For example, suppose the precision of the AI-based malware classification algorithm is 0.95. This is a fairly high number, but it is ambiguous to use as a final step in malware detection. If this method is used as the final stage of malware detection, even under ideal circumstances, 5% of malware bypasses this method and is not completely free from political and economic damage.

Ironically, this article is not an argument that AI-powered technologies should not be used. This is because the rule-based detection method in the final stage can be applied to data or network traffic whose volume has been drastically reduced after being primarily filtered through AI-based technology. It is labor-intensive to analyze the S/W that is mass-produced every day, and it is not even necessary to do so. This is an example of the effective use of AI technology in consideration of the domain characteristics of the information security field. Although the purpose of this study is not to optimize the performance of the FL technique, considering the accuracy of 91%, it is inappropriate to be used alone, and it should be cooperated with the strict malware detection technique in the final stage, possibly installed in clients.

### 6.2. Countermeasures against Adversarial Attack

In FL, model training performed on a central cloud data server is distributed to clients, making them more vulnerable to adversarial attacks by extending the target of adversarial attacks on clients. One known countermeasure against adversarial attacks is to train deep learning models based on training data containing artificially generated adversarial examples (AEs), which is called adversarial training. However, creating an effective AE on the client side renders a new challenge in FL, as the client’s computing power is limited and global data visibility is not secured. This ultimately leads to the problem of efficient distribution of computing and is being actively studied [36].

## 7. Conclusions

Our experiments show that FedAvg can be adopted to make federated learning feasible and efficient while preserving the client’s privacy. The impacts of client availability, local computing costs, and communication overhead between the center and local were investigated in depth. While federated learning offers many practical privacy advantages in real mobile networks, problems such as the algorithmic distribution of computational resources for adversarial training or differential computations are extended to FL-based distributed environments, opening up interesting and worthy future research directions.

## Figures and Tables

**Figure 1 sensors-23-02198-f001:**
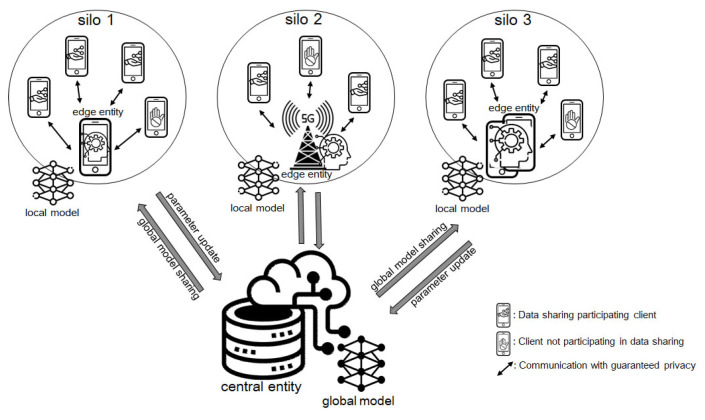
The cross-silo FL model assumed in this paper consists of one central entity and a plurality of edge entities. The edge entity is a form of a coalition between the client(s) and the base station, and a representative part of the component terminals communicates directly with the central entity to participate in global model learning.

**Figure 2 sensors-23-02198-f002:**
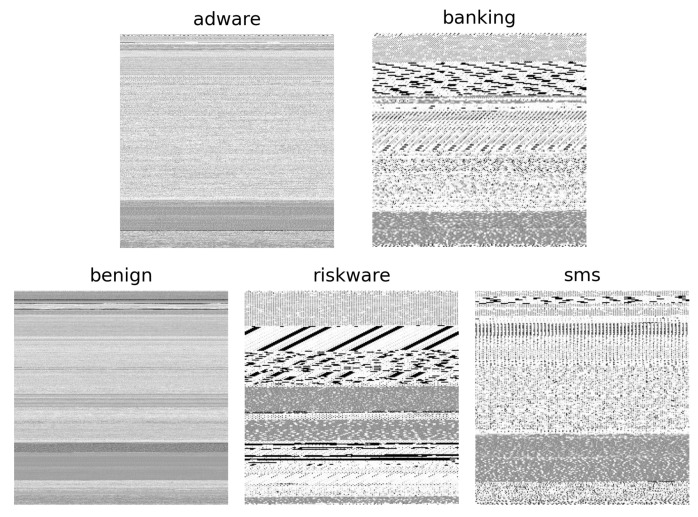
This is the result of converting APK files of five classes (benign and four malignant classes; adware, banking, riskware, and sms) constituting the CICMaldroid dataset into image files using Stream Order (SO). A single image was randomly selected for each class. If an image is chosen from the entire dataset, it is almost impossible to visually identify which class the selected image belongs to.

**Figure 3 sensors-23-02198-f003:**
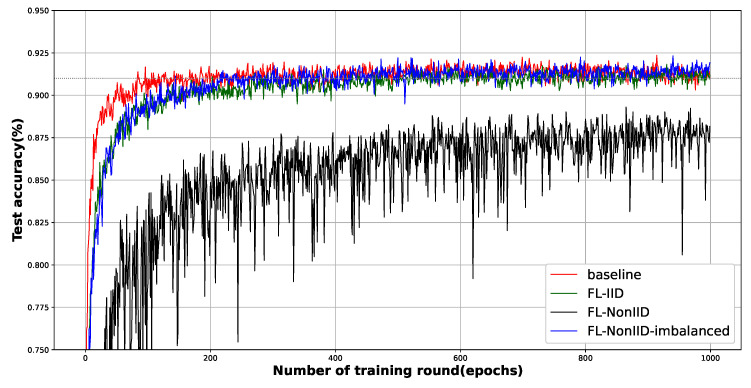
The figure shows the test accuracy of four schemes, i.e., the centralized CNN-based deep learning model (baseline) and the CNN-based deep learning model with FL on three data distributions—IID, non-IID, and non-IID imbalanced. Training rounds ran up to 1000 epochs. The horizontal gray dotted line indicates 0.91%.

**Figure 4 sensors-23-02198-f004:**
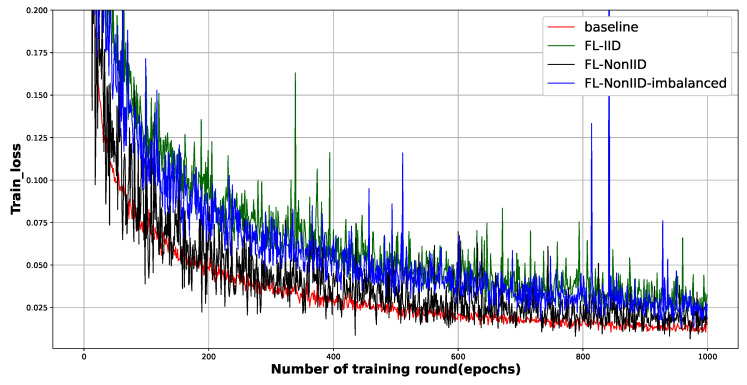
The figure shows the training loss of four schemes, i.e., the centralized CNN-based deep learning model (baseline) and the CNN-based deep learning model with FL on three data distributions—IID, non-IID, and non-IID imbalanced. Training rounds ran up to 1000 epochs.

**Figure 5 sensors-23-02198-f005:**
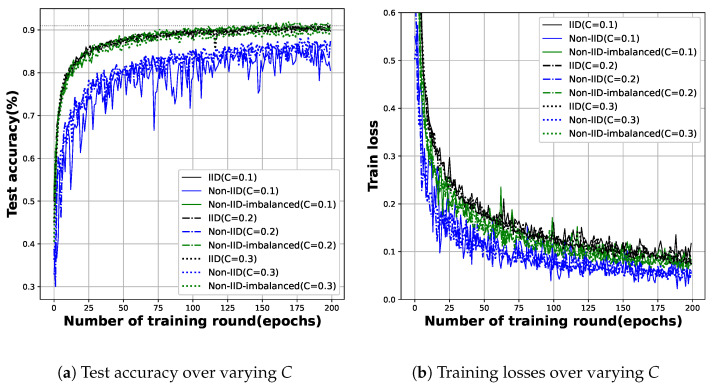
This figure shows the test accuracy and training loss of FL as the edge entity’s availability (*C*) varies under three different data distributions.

**Figure 6 sensors-23-02198-f006:**
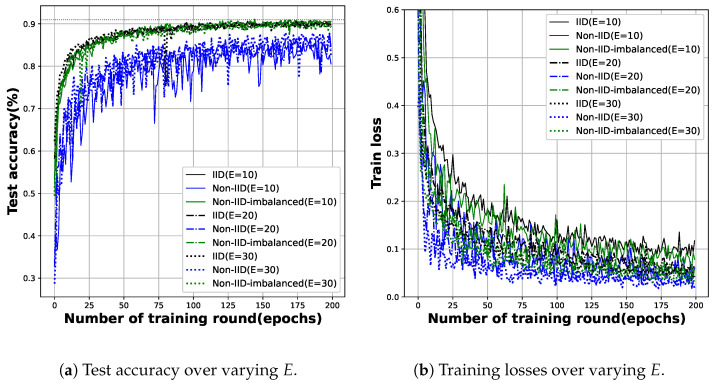
This figure shows the test accuracy and training loss of FL as the number of local training rounds (*E*) varies under three different data distributions.

**Table 1 sensors-23-02198-t001:** Parameters used in our experiments.

Notation	Values	Meaning	Remarks
*C*	0.1, 0.2∼1	The fraction of edge entities	
*E*	10, 20, 30	The number of local epochs	
*K*	100	Number of users	Not controlled
*B*	10	Local batch size	Not controlled
N/A	0.01	learning rate	Not controlled
N/A	0.5	SGD momentum	Not controlled

**Table 2 sensors-23-02198-t002:** This table presents the number of training rounds, training time, and speed-ups compared to the baseline for the four schemes covered in this paper to achieve a target test accuracy of 0.91. Among the four schemes, FL on non-IID data never achieves a test accuracy of 0.91 even for 1000 rounds and converges around 0.87, setting the target test accuracy for this method to be 0.87.

Scheme	*C*	*E*	# of Training Round	Training Time	Speed-Ups
Baseline (FedSGD)	-	-	84	9 h 6 m 20.81 s	-
FL(IID)	0.1	10	203	5 h 9 m 32.50 s	1.76×
	0.2		131	5 h 53 m 16.95 s	1.54×
	0.3		150	10 h 6 m 46.96 s	0.90×
	0.1	10	203	5 h 9 m 32.50 s	1.76×
		20	133	2 h 59 m 20.28 s	3.04×
		30	95	2 h 8 m 05.92 s	4.27×
FL(Non-IID)	0.1	10	253	5 h 58 m 12.25 s	1.52×
	0.2		189	8 h 29 m 41.85 s	1.07×
	0.3		144	9 h 42 m 30.69 s	0.93×
	0.1	10	253	5 h 58 m 12.25 s	1.52×
		20	195	4 h 22 m 56.35 s	2.08×
		30	119	2 h 40 m 27.62 s	3.40×
FL(Non-IID-imbalanced)	0.1	10	146	3 h 16 m 52.04 s	2.77×
	0.2		129	5 h 47 m 53.33 s	1.57×
	0.3		189	12 h 44 m 32.78 s	0.71×
	0.1	10	146	3 h 16 m 52.04 s	2.77×
		20	124	2 h 47 m 12.14 s	3.27×
		30	151	3 h 23 m 36.56 s	2.68×

## Data Availability

https://www.unb.ca/cic/datasets/maldroid-2020.html.

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
