# Peer review of "Distributed Detection of Malicious Android Apps While Preserving Privacy Using Federated Learning"

_sensors, 2023, doi:10.3390/s23042198_

Round 1
Reviewer 1 Report
1.The description of the proposed work is not prominent in this article, and most of the description in Section 3 is a combination of the others' work (Ref. 15 and 29), please revise and add to it.
2.The article is titled "Distributed Detection of Malicious Android Apps while Preserving Privacy using Federated Learning", but the article lacks a detailed analysis of the privacy preserving.
3.The description of the experimental hardware (CPU/GPU/Memory) and simulation software (Software tools/Machine learning library/Other libraries) is missing in Section IV of the article, please add them.
Author Response
Response to Reviewer 1 Comments
Point 1: The description of the proposed work is not prominent in this article, and most of the description in Section 3 is a combination of the others' work (Ref. 15 and 29), please revise and add to it.
Response 1: As raised by the reviewer 1, we agree that the description of the proposed work is not prominent. To address this, we revised the second to last paragraph of the Section 1. Since the mathematical formula is not directly related to the development of the manuscript and has already been presented in numerous papers, we removed the mathematical formula and reorganized Section 3, leaving a description of the parameter update process. These revisions made our proposed work clearer.
Point 2: The article is titled "Distributed Detection of Malicious Android Apps while Preserving Privacy using Federated Learning", but the article lacks a detailed analysis of the privacy preserving.
Response 2: As raised by the reviewer 1, we added detailed analysis of the privacy preserving using local differential privacy (LDP) at the end of Section 3.1.
Point 3: The description of the experimental hardware (CPU/GPU/Memory) and simulation software (Software tools/Machine learning library/Other libraries) is missing in Section IV of the article, please add them.
.Response 3: In the previous submission, it was difficult to find because this content was briefly described before describing the experimental results in terms of efficiency in Section 4.2. As suggested by the reviewer, this content was moved to Section 4.1 and reinforced.

Reviewer 2 Report
This paper studies the various methods of detecting malicious codes, and reclassifies the related work from two aspects, signature-based and behavior-based. Finally, a novel Android malicious app detection method is proposed by introducing federated learning technology.
The comments are as follows:
1. The related work of federation optimization needs further supplement.
2. Please explain the characteristics of the proposed model, especially in solving the leakage of personal information in the coalition between the edge entity and the client.
3. The explanation of the imaging technique for malicious Android app samples in the form of APKs can be described more clearly.
4. Please explain in detail the purpose of using FedSGD as the baseline.
5. The advantages of the proposed model are not fully described.
Author Response
Response to Reviewer 2 Comments
This paper studies the various methods of detecting malicious codes, and reclassifies the related work from two aspects, signature-based and behavior-based. Finally, a novel Android malicious app detection method is proposed by introducing federated learning technology.
The comments are as follows:
Point 1: The related work of federation optimization needs further supplement.
Response 1: As suggested by reviewer 2, we additionally cited the following key papers on federated optimization. Section 2 was reorganized to improve readability.
[28] Li, X.; Huang, K.; Yang,W.;Wang, S.; Zhang, Z. On the convergence of fedavg on non-iid data. arXiv preprint arXiv:1907.02189 2019.
[29] Arivazhagan, M.G.; Aggarwal, V.; Singh, A.K.; Choudhary, S. Federated learning with personalization layers. arXiv preprint 559 arXiv:1912.00818 2019.
[30] Li, T.; Sahu, A.K.; Zaheer, M.; Sanjabi, M.; Talwalkar, A.; Smith, V. Federated optimization in heterogeneous networks. Proceedings of Machine learning and systems 2020, 2, 429–450.
Point 2: Please explain the characteristics of the proposed model, especially in solving the leakage of personal information in the coalition between the edge entity and the client.
Response 2: We added detailed analysis of the privacy preserving using local differential privacy (LDP) at the end of Section 3.1. And Section 3 has been reorganized/rewritten to focus on describing the proposed model.
Point 3: The explanation of the imaging technique for malicious Android app samples in the form of APKs can be described more clearly.
Response 3: As the reviewer 2 suggested, the method of imaging the APK file was rewritten to improve readability in Section 4.1.1.
Point 4: Please explain in detail the purpose of using FedSGD as the baseline.
Response 4: In reorganizing Section 3, we have moved the paragraph describing the baseline from Section 4 to the end of Section 3.3. And it further explains why the FedSGD is called the baseline.
Point 5: The advantages of the proposed model are not fully described.
Response 5: To address this, we revised the second to last paragraph of the Section 1. Since the mathematical formula is not directly related to the development of the manuscript and has already been presented in numerous papers, we removed the mathematical formula and reorganized Section 3, leaving a description of the parameter update process. These revisions made our proposed work clearer.

Round 2
Reviewer 1 Report
The authors have addressed all my concerns.